# Lanthanides-Based Nanoparticles Conjugated with Rose Bengal for FRET-Mediated X-Ray-Induced PDT

**DOI:** 10.3390/ph18050672

**Published:** 2025-05-01

**Authors:** Batoul Dhaini, Joël Daouk, Hervé Schohn, Philippe Arnoux, Valérie Jouan-Hureaux, Albert Moussaron, Agnès Hagege, Mathilde Achard, Samir Acherar, Tayssir Hamieh, Céline Frochot

**Affiliations:** 1Université de Lorraine, CNRS, LRGP, F-54000 Nancy, France; batoul.dhaini@outlook.com (B.D.); philippe.arnoux@univ-lorraine.fr (P.A.); albert.moussaron@hotmail.fr (A.M.); 2Laboratory of Materials, Catalysis, Environment and Analytical Methods (MCEMA), Faculty of Sciences I, Lebanese University, Beirut P.O. Box 6573/1, Lebanon; t.hamieh@maastrichtuniversity.nl; 3Université de Lorraine, CNRS, CRAN, F-54000 Nancy, France; joel.daouk@univ-lorraine.fr (J.D.); herve.schohn@univ-lorraine.fr (H.S.); valerie.jouan-hureaux@univ-lorraine.fr (V.J.-H.); 4Université Claude Bernard Lyon 1, CNRS, ISA, UMR 5280, F-69100 Villeurbanne, France; agnes.hagege@univ-lyon1.fr; 5Université de Lorraine, CNRS, LCPM, F-54000 Nancy, France; mathilde.achard@univ-lorraine.fr (M.A.); samir.acherar@univ-lorraine.fr (S.A.); 6Faculty of Science and Engineering, Maastricht University, 6211 LH Maastricht, The Netherlands

**Keywords:** fluorescence resonance energy transfer (FRET), photodynamic therapy induced by X ray (X-PDT), Rose Bengal (RB), terbium, gadolinium, peptide, glioblastoma, AGuIX, Neuropilin 1 (NRP-1)

## Abstract

In order to find a good candidate for Förster Resonance Energy Transfer (FRET)-mediated X-ray-induced photodynamic therapy (X-PDT) for the treatment of cancer, lanthanide (Ln)-based AGuIX nanoparticles (NPs) conjugated with Rose Bengal (RB) as a photosensitizer (PS) were synthesized. X-PDT overcomes the problem of the poor penetration of visible light into tissues, which limits the efficacy of PDT in the treatment of deep-seated tumors. It is essential to optimize FRET efficiency by maximizing the overlap integral between donor emission and acceptor absorption and lengthening the duration of the donor emission. In this study, we optimized energy transfer between a scintillator (Sc) as a donor and a PS as an acceptor. Terbium (Tb) and Gadolinium (Gd) as Scs and Rose RB as a PS were chosen. The study of energy transfer between Tb, Gd and RB in solution and chelated on AGuIX NPs proved to be FRET-like. RB was conjugated directly onto AGuIX NPs (i.e., AGuIX Ln@RB), and the use of a spacer arm (i.e., AGuIX Ln@spacer arm-RB) increased FRET efficiency. Singlet oxygen production by these NPs was observed under UV–visible illumination and X-ray irradiation. The in vitro bioassay demonstrated 52% cell death of U-251MG derived from human malignant glioblastoma multiforme at a concentration of 1 μM RB after illumination and irradiation (2 Gy, 320 kV, 10 mA, 3 Gy/min at 47 cm). In addition, the RB-coupled NRP-1-targeting peptide (i.e., K(RB)DKPPR) was conjugated onto AGuIX NPs by a thiol-maleimide click chemistry reaction, and an affinity in the nM range was observed.

## 1. Introduction

Photodynamic therapy (PDT) for cancer appears to be an excellent candidate for treating glioblastoma and other types of solid cancer. PDT involves injecting a photoactivatable molecule called a photosensitizer (PS), then exciting it in the tumor area after a time interval known as the “drug–light interval”. This excitation leads to the production of reactive oxygen species (ROS), notably singlet oxygen (^1^O_2_). PDT offers several advantages over other types of treatment, such as non-toxicity of PS in the dark, absence of treatment resistance and few side effects. PDT, on the other hand, faces several obstacles: lack of PS selectivity, limited light penetration into deep tissues and lack of oxygenation. In this paper, we focus on the use of X-ray, which penetrates deep into the tissues, to excite nanoparticles (NPs). Indeed, in recent years, photodynamic X-ray excitation therapy (X-PDT), which uses penetrating X-rays as an external excitation source and luminescent X-ray-excited nanoparticles as an energy-transfer medium to indirectly excite the PS, has developed considerably to address the problem of insufficient tissue penetration depth in particular. Recent reviews describe the various nanoparticles used for these purposes [1,2]. The advantage of scintillating materials composed of atoms with a high atomic number (high Z) such as terbium (Tb) or gadolinium (Gd) is that they are able to fluoresce under the excitation of high-energy radiation. This fluorescence is then absorbed by the PS to continue the PDT process [3]. Moreover, NPs have the ability to target tumors passively, named thereafter passive targeting, through the enhanced permeability effect (EPR). They have also been developed for active targeting by the graft of peptides which bind specifically to receptors overexpressed on membranes of cancer cells or neovascularization [4].

In biomedicine and pharmacology, the physiological mechanisms of DNAs [5], micro RNAs [6], and enzymes [7] are detected with fluorescent molecules as biomarkers. The mechanism of these biomarkers is based on the principle of Förster Resonance Energy Transfer (FRET) between two fluorophores. In medicine and environmental research, FRET-based biomarkers are also used for toxicity [8] and quality control studies, such as for the detection of organophosphate pesticides.

The energy transfer process between a donor and an acceptor can be radiative or non-radiative (Dexter versus FRET). Radiative energy transfer is based on the emission of a photon by a donor which is then absorbed by an acceptor. This energy transfer takes place without interaction between donor and acceptor and is therefore long-distance. Non-radiative energy transfer occurs via an interaction between donor and acceptor, either a Coulomb dipole/dipole interaction at medium distances between 1 and 10 nm (Förster mechanism) or by a multipole interaction at shorter distances (Dexter mechanism) (Figure 1).

Numerous studies have been carried out to improve PDT-X. Some research has focused on strategies to enhance luminescence. Methods are also being developed to efficiently load PSs into NPs and improve their solubility in aqueous media. On the other hand, it is important to adjust the distance between the donor and acceptor to increase the energy transfer efficiency [2]. Other researchers are trying to find a compromise between the therapeutic index and the ability for in-depth treatment without increasing the toxicity [9]. Research indicates that the use of NPs in PDT-X is very promising [10], especially NPs synthesized with rare-earth metals [1].

In this paper, we study energy transfer between Terbium (Tb), Gadolinium (Gd) as a donor and Rose Bengal (RB) as an acceptor. Gd is used in therapy and imaging techniques as it significantly increase the contrast intensity of Magnetic Resonance Imaging (MRI) and may also have very promising effects as a radiosensitizer [11]. Tb, unlike Gd, which has a single fluorescence emission peak, presents four visible fluorescence emission peaks that overlap with the UV–visible absorption spectrum of RB. The interest in lanthanides (Lns) for such application is due to their 4f orbital [12,13,14,15,16,17,18,19,20,21,22,23].

RB is a xanthene PS [22] derived from fluorescein (Figure 1a) with interesting photophysical and sonosensitive properties [23,24]. For example, Nonaka et al. [25] used sonodynamic therapy (SDT) with RB and focused ultra-sound to treat experimental intracranial glioma in rats. A further application of sono-activated RB was reported by Nakonechny et al. [26] who demonstrated that it was possible to eradicate Gram-positive and Gram-negative bacteria by applying SDT using activated RB in vitro. RB photoactivation can be used for external application on the body, such as wound sealing or corneal crosslinking, whereas the use of sono-activated RB could be further explored for cancer treatment. In water, the maximum absorption wavelength is 545 nm (Figure 1b). In ethanol, its singlet oxygen quantum yield (Φ_Δ_) is 0.68 [27]. It also exhibits significant fluorescence properties (Figure 1b), with a fluorescence quantum yield (Φ_f_) of 0.11 in ethanol [28].

RB can be used in PDT due to its production of ^1^O_2_ after light excitation and gives excellent results in anti-bacterial and anti-cancer PDT [4]. RB is not selective for cancer cells. One of the keys to overcoming this problem is to couple RB to nanoparticles (NPs), enabling passive targeting of cancer cells via the enhanced permeability and retention (EPR) effect. In the literature, several types of NPs have been linked to RB for PDT, such as silica NPs [29], organic NPs, nanogels [30], nanocomplexes [31], hybrid NPs [32], and MOFs [33]. The covalent coupling of RB to NPs is considered more efficient than encapsulation [4].

In this study, AGuIX NPs were selected because they offer several advantages (Figure 2). Firstly, AGuIX NPs have an average hydrodynamic diameter of about 3.5 ± 1.0 nm and a mass of about 10 kDa, enabling simple renal elimination [34]. AGuIX NPs are composed of a polysiloxane matrix and concentrate a high number of Gd atoms (around 15). AGuIX NPs are currently being evaluated in a Phase 2 clinical trial in combination with the standard of care for several indications (glioblastoma, brain metastasis, lung and pancreatic cancer), while Phase I clinical trials results about brain metastasis and cervical cancer indications have been published [35,36].

In the nano AGuIX platform tested, firstly, DOTA was chelated with Gd or Tb, and secondly, a targeted peptide recognizing neuropilin 1 (NRP-1) coupled to RB (i.e., K(RB)DKPPR) was conjugated to the surface of AGuIX NPs [35]. Interestingly, the peptide KDKPRR alone has been shown to behave in an affinity of the order of μM for NRP-1 [37]. In K(RB)DKPPR, conjugation via a maleimide function enhanced the affinity constant to the order of nM. Finally, NPs were tested in vitro on U-251 MG cells, derived from a human glioblastoma multiforme, using an anchorage-dependent clonogenic assay. X-ray irradiation (320 kV, 10 mA, 3 Gy/min) of AGuIX Tb NPs coupled to RB led to 48% cell survival at a concentration equivalent of 1 μM of RB. Similar results were obtained when cell were exposed to NPs doped with Tb and the RB-peptide K(RB)DKPPR.

## 2. Results and Discussion

Energy transfer between AGuIX (Tb) or AGuIX (Gd) and RB was first evaluated in a solution under light excitation.

### 2.1. Energy Transfer Between Tb, Gd and RB

#### 2.1.1. Photophysical Properties of Tb (TbCl_3_)

In order to detect a potential energy transfer between AGuIX (Tb) or AGuIX (Gd) and RB, model molecules (TbCl_3_ or GdCl_3_) were first studied.

The UV–visible absorption spectrum of Tb (TbCl_3_) in water shows a maximum absorption peak at 219 nm (Figure 3a). However, after excitation of Tb at 219 nm, no luminescence emission could be detected. Figure 3b shows the excitation spectrum of TbCl_3_ for fluorescence emission at 545 nm. The first excitation peak after 219 nm is 72 nm, but there is still no luminescence present. The next excitation pick is 351 nm, so we decided to excite at 351 nm. After excitation at 351 nm, the fluorescence emission spectrum of Tb (TbCl_3_) showed the four characteristic peaks of Tb at 488 nm, 545 nm, 585 nm and 620 nm (Figure 3c). These Tb fluorescence emission peaks correspond to the electronic transitions between the ^5^D_4_ and ^7^F_6_, ^7^F_5_, ^7^F_4_ and ^7^F_3_ energy levels, respectively.

#### 2.1.2. Photophysical Properties of Gd (GdCl_3_)

The UV–visible absorption spectrum of Gd (GdCl_3_) in water shows a maximum absorption peak at 273 nm (Figure 4a). After excitation of Gd at 273 nm, a large fluorescence emission peak at 313 nm is observed (Figure 4b) after a delay of 50 μs. This fluorescence emission peak corresponds to the energy difference between the ^6^P_J_ and ^6^S_7/2_ energy levels. *Among the Ln elements*, Gd is the only one to have a too-high first energy state, which justifies the single narrow emission peak.

#### 2.1.3. Energy Transfer Between Terbium (TbCl_3_), Gadolinium (GdCl_3_) and RB in Water

The overlap integral J_(λ)_ and the Förster radius R_0_ were calculated from Equations (1) and (2), respectively, in the long recovery wavelength region. For the TbCl_3_/RB and GdCl_3_/RB pairs, we found a J_(λ)_ value of 4.36 × 10^15^ M^−1^·nm^4^·cm^−1^ and 2.72 × 10^14^ M^−1^·nm^4^·cm^−1^, respectively, and an R_0_ value of 4.33 nm and 2.73 nm. These J_(λ)_ values, together with the R_0_ < 10 nm, suggest the possibility of FRET. To substantiate this hypothesis, we investigated the variation in luminescence intensity and lifetime of Tb and Gd as a function of the RB concentration, with a fixed donor concentration ([Tb^3+^] and [Gd^3+^] = 10 mM).

Figure 5 shows (a,b) the spectral overlap J_(λ)_ between the Ln emission and RB absorption (Ln = Tb and Gb, respectively), (c,d) a decrease in Ln luminescence intensity after RB (Ln = Tb and Gb, respectively), (e,f) a decrease in Ln luminescence lifetime after RB addition (Ln = Tb and Gb, respectively), and (g,h) I_0_/I = f([RB]) and τ_0_/τ = f([RB]) in water for Ln (λ_exc_ (Ln) = 351 nm, 50 μs) (Ln = Tb and Gb, respectively).

The decrease in luminescence intensity and lifetime for Tb and Gd is consistent with non-radiative energy transfer.

It is possible to compare the two GdCl_3_/RB and TbCl_3_/RB pairs. The spectral overlap between Tb and RB is 10 times higher than that of Gd and RB, leading to better energy transfer for the TbCl_3_/RB pair than that of GdCl_3_/RB. It is possible to evaluate, according to Equation (3), the simplified energy transfer efficiency (E) using the luminescence lifetimes without and with quenching. E is approximately 65% for the TbCl_3_/RB pair and 42% for the GdCl_3_/RB pair.

#### 2.1.4. Energy Transfer Between AGuIX Tb, AGuIX Gd and RB in Water

Energy transfer between AGuIX Tb or AGuIX Gd and RB was evaluated in water. For the AGuIX Tb/RB pair, a spectral overlap value J_(λ)_ of 1.87 × 10^15^ M^−1^·nm^4^·cm^−1^ and a Förster radius R_0_ of 3.76 nm were obtained. The R_0_ value was of the same order of magnitude as that of TbCl_3_/RB (R_0_ = 4.33 nm). This spectral overlap was greater than that of the AGuIX Gd/RB pair, which had a J_(λ)_ value of 5.60 × 10^14^ M^−1^·nm^4^·cm^−1^ and an R_0_ value of 3.08 nm. These values were of the same order of magnitude as that of GdCl_3_/RB (J_(λ)_ = 2.72 × 10^14^ M^−1^·nm^4^·cm^−1^ and R_0_ = 2.73 nm). For both AGuIX Tb/RB and AGuIX Gd/RB, energy transfer was non-radiative with dynamic and static inhibitions. We could calculate kq = 0.57 × 10^8^ M^−1^·s^−1^ for AGuIX Gd/RB, given that Ksv = 11.40 × 10^4^ M^−1^. In the case of AGuIX Tb/RB, Ksv = 4.9632 × 10^4^ M^−1^ leading to kq = 0.04932 × 10^8^ M^−1^·s^−1^.

The same excitation wavelengths as TbCl_3_ and GdCl_3_ (λ_exc_ of 351 and 273 nm, respectively) with a delay of 50 μs in water were used for AGuIX Tb and AGuIX Gd.

The luminescence emission spectra of AGuIX Tb and AGuIX Gd alone were recorded, as well as in the presence of RB (Figure 6). After a delay of 50 μs, the emission of AGuIX Tb and AGuIX Gd (black) in the presence of RB (red) decreases and a fluorescence emission from RB appears between 550 nm and 600 nm. These results support energy transfer between AGuIX NPs and RB.

In short, the FRET efficiencies between different couples were calculated according to Equation (3) and are presented in Table 1.

As expected from the J overlap, the FRET efficiency is more efficient between AGuIX Tb and RB than between AGuIX Gd and RB.

### 2.2. Passive Targeting

Since an energy transfer between AGuIX (Tb) or AGuIX (Gd) and RB was observed in solution under light excitation, RB was covalently coupled to Ln-based AGuIX NPs in order to reduce the distance between the lanthanide and the PS to increase the FRET efficiency.

#### 2.2.1. Covalent Binding Between Ln-Based AGuIX NPs and RB

The results in the solution encourage us to covalently couple RB to Ln-based AGuIX NPs without a spacer arm (i.e., AGuIX Ln@RB) and with a six-carbon spacer arm (i.e., AGuIX Ln@spacer arm-RB) via an ester bond (hexanoic acid, AGuIX Ln@HA-RB) or an amide bond (aminohexanoic acid, AGuIX Ln@AhxRB). The AGuIX Ln@RB and AGuIX Ln@spacer arm-RB (see synthesis protocol in Appendix A) showed an increase in the zeta potential (ζ) in absolute values compared with AGuIX Ln alone. Covalent coupling of RB with or without a spacer arm increased the NPs’ stability. The increase in negative charge reflects the fact that at pH 7.2, RB phenolic groups can be deprotonated to phenolate anions. ζ values obtained by Dynamic Light Scattering (DLS) and size values obtained by TDA-ICP/MS for AGuIX Ln, AGuIX Ln@RB and AGuIX Ln@spacer arm-RB are shown in Table 2. As can be seen for Tb-based NPs, the TDA analysis shows that AGuIX Tb @RB, AGuIX Tb@HA-RB and AGuIX Tb@Ahx-RB exhibit a higher hydrodynamic diameter, which reflects the efficiency of the coupling. Moreover, the analysis reveals the presence of two populations in these samples, with a main population which can be attributed to the functionalized NP, while the other remaining population might be attributed to a slight hydrolysis of the NP (see Appendix A).

Figure 7a,c shows the UV–visible absorption spectra of RB, AGuIX Ln, AGuIX Ln@RB and the AGuIX Ln@spacer arm-RB in water. A bathochromic shift is observed compared to the RB peak alone, with a small broadening between 460 nm and 620 nm. The strong UV broadening in Figure 7c is probably due to the coupling of RB with AGuIX Gd. This result favors covalent coupling of RB in AGuIX Gd@RB and the AGuIX Gd@spacer arm-RB. Figure 7b,d shows the luminescence emission spectra of AGuIX Ln, AGuIX Ln@RB and the AGuIX Ln@spacer arm-RB in water at the same Tb/RB molar ratio. This energy transfer appears to be greater in the presence of a spacer arm (HA and Ahx). Both arms have the same number of carbons, and we have assumed that the distance between the donor and acceptor is not radically different.

Table 3 summarizes the photophysical properties of RB, AGuIX Ln, AGuIX Ln@RB and AGuIX Ln@spacer arm-RB.

The Ln luminescence lifetime is recorded after excitation at 545 nm for Tb and 313 nm for Gd. The fluorescence lifetime of RB is recorded after excitation at 470 nm ($\tau_{f_{470}}$). ^1^O_2_ emission spectra are obtained after excitation at 558 nm (maximum absorption wavelength of RB) but also after excitation at 351 nm for Tb and 273 nm for Gd.

An increase in $\tau_{f_{470}}$ once RB was coupled to the NPs by a factor of around 10 was observed (Table 2). With regard to the fluorescence and ^1^O_2_ quantum yields recorded at 558 nm ($\Phi_{f_{558}}$ and $\Phi_{\Delta_{558}}$), no significant variation was observed between RB alone vs. AGuIX Ln@RB and AGuIX Ln@spacer arm-RB. These results highlighted the conservation of photophysical properties of RB coupled to AGuIX Ln NPs. The ^1^O_2_ quantum yields recorded at 351 nm for Tb ($\Phi_{\Delta_{351}(Tb)}$) and at 273 nm for Gd ($\Phi_{\Delta_{273}(Gd)}$) were similar for AGuIX Ln@RB and AGuIX Ln@spacer arm-RB. Importantly, the values of ^1^O_2_ quantum yields after excitation at 273 nm for AGuIX Gd and 351 nm for AGuIX Tb are half of those recorded at 558 nm. In conclusion, ^1^O_2_ production is greater when excitation is localized to RB (PDT effect) than to Ln (X-PDT effect).

We then assessed whether ^1^O_2_ generation is achieved under a range of X-ray doses (320 kV/10 mA). To demonstrate ^1^O_2_ formation under a range of X-ray doses (320 kV/10 mA), the singlet oxygen sensor green (SOSG) probe was used. As shown in Figure 8, an increase in SOSG fluorescence signal in the presence of AGuIX Ln@RB and AGuIX Ln@spacer arm-RB was seen, supporting the production of ^1^O_2_. In contrast, no ^1^O_2_ was generated with AGuIX Ln or RB alone. Addition of the ^1^O_2_ quencher NaN_3_ decreased the SOSG signal, demonstrating that ^1^O_2_ generation is due to energy transfer between Ln and RB. Furthermore, AGuIX Ln or RB alone do not produce ^1^O_2_ under X-ray excitation. Tb is more efficient as an energy transfer donor than Gd, as shown by the best spectral overlap between the RB absorption spectrum and the Tb emission spectrum compared with the Gd emission spectrum (Figure 5). AGuIX Tb@HA-RB and AGuIX Tb@-RB show the highest ^1^O_2_ production at different X-ray doses (320 kV, 10 mA).

#### 2.2.2. Cell Clonogenic Assays

The characterized NPs were then used to study their impact on cell growth using cell anchorage-clonogenic assays (Figure 9). U-251 MG cells pre-treated with NPs for 24 h were exposed to X-ray irradiation (2 Gy, 320 kV, 10 mA, 3 Gy/min at 47 cm) and cell clones were quantified 7 days after X-ray exposition (Figure 9) based on previous work evaluating AGuIX@Tb-Porphyrin, as a PS [38]. Whatever the composition of the Ln-based AGuIX tested, cell growth was reduced. Interestingly, the number of cell clones obtained was reduced to 68% when cell were treated in the presence of AGuIX Gd@HA-RB and, surprisingly, to 49% with AGuIX Gd@RB, but the results were not significant. The observed decrease in cell growth after X-ray exposure, in terms of cell death mechanisms, was not analyzed. However, cells could undergo diverse cell death mechanisms such as autophagy, apoptosis, necrosis or mitotic catastrophe. The latter is considered to be the major mechanism after cell exposure to ionisation [39].

### 2.3. Active Targeting

For active targeting, we decided to use an NRP-1-targeting peptide that we coupled to RB (i.e., K(RB)DKPPR. The NRP-1-targeting peptide KDKPPR has already been described by our team with micromolar affinity [40]. NRP-1 is overexpressed in angiogenic endothelial cells and in certain tumors such as glioblastoma and breast cancer [41]. K(RB)DKPPR was conjugated to AGuIX Ln NPs by a thiol-maleimide click chemistry reaction to obtain AGuIX Ln@Mal-K(RB)DKPPR NPs. A maleimide arm was added to the *N*-terminus of K(RB)DKPPR (i.e., Mal-K(RB)DKPPR), and amino groups on the surface of AGuIX Ln NPs were converted to thiol groups (see Appendix A). Figure 10 shows (a) the UV–visible absorption and (b) luminescence spectra of AGuIX Ln, AGuIX Ln@Mal-K(RB)DKPPR and control AGuIX Ln@Mal-K(RB).

In all cases, the coupling of RB to NPs resulted in a bathochromic shift in its absorption peak (Figure 10a). This finding indicated the successful covalent coupling of RB derivatives to AGuIX Ln NPs. Furthermore, a decrease in luminescence intensity, after excitation with a delay of 50 μs, of all RB-coupled AGuIX Ln compared to AGuIX Ln alone highlights the presence of FRET (Figure 10b). The ^1^O_2_ and fluorescence quantum yields, RB fluorescence, Ln luminescence lifetimes, zeta potential and size are detailed in Table 4.

The size of all NPs was measured by TDA-ICP-MS (Taylor Dispersion Analysis coupled to Inductively Coupled Plasma Mass Spectrometry). The size of all NPs was less than 40 nm (Table 4) and two size populations could be observed. One population is composed of AGuIX and the other of a set of AGuIX. The difference between the two populations can be attributed to a weak stacking phenomenon causing aggregation in water.

Using a similar experimental approach as in Figure 8, we tested whether each NP produced ^1^O_2_ under X-ray irradiation (320 kV/10 mA). An increase in SOSG fluorescence signal was observed confirming ^1^O_2_ production, which was inhibited by the addition of NaN_3_ in the reaction mixture (Figure 11). To demonstrate the potential interest of the AGuIX Ln@Mal-(K(RB)DKPPR NPs, we assessed the affinity constant of each NP for NRP-1 (Figure 12). KDKPPR alone has a binding affinity of around 1 μM [41]. Interestingly, the NRP-1 affinity of AGuIX Ln@Mal-K(RB)DKPPR NPs displayed a 1000-fold increase compared to the KDKPPR peptide alone. This was estimated at 6 and 0.5 nM or AGuIX Gd@Mal-K(RB)DKPPR and AGuIX Tb@Mal-K(RB)DKPPR, respectively, which might be due to a difference in dispersion in water. In this concentration range, the IC50 could not be assessed forAGuIX Gd@Mal-K(RB) or AGuIX Tb@Mal-K(RB), showing the lower affinity of these nanoparticles for NRP-1.

Based on the results obtained, anchorage-dependent clonogenic assays were performed. U251 MG cells were pre-treated in the presence of AGuIX Ln@Mal-K(RB)DKPPR or AGuIX Ln@Mal-K(RB) and irradiated with a dose of 2 Gy (320 kV, 10 mA, 3 Gy/min at 47 cm). As shown in Figure 13, the formation of cell clones was lower when cells were treated with AGuIX Ln@Mal-K(RB)DKPPR, compared to one with AGuIX Ln@Mal-K(RB). The strongest inhibition of cell growth was obtained when cells were exposed to AGuIX Tb@Mal-K(RB)DKPPR (57% decrease, *p* = 0.02). The absence of the DKPPR peptide on the NPs had no impact on cell survival, in contrast to the results obtained with NPs containing the peptide.

## 3. Materials and Methods

### 3.1. Chemicals and Materials

#### 3.1.1. Chemicals

Ultrapure water (Milli–Q, ρ >18 MΩ · cm) was used in all experiments and all purchased chemicals were used without further purification. Dichloromethane (DCM), dimethylformamide (DMF), ethanol (EtOH), methanol (MeOH), chloroform (CHCl_3_), acetonitrile (ACN), dimehylsufoxide (DMSO), RB sodium salt (95%), 6-bromohexanoic acid (HA spacer arm, 97%), 6-aminohexanoic acid (Ahx spacer arm, 98%), 6-maleimidohexanoic acid (Mal, 98%), 2-iminothiolane hydrochloride (Traut’s reagent, 98%), trifluoroacetic acid (TFA, 99%), acetic anhydride (98%), piperidine (99%), triethylamine (99%), *N*-hydroxysuccinimide (NHS, 98%), *N*,*N′*-dicyclohexylcarbodiimide hydrochloride (EDC.HCl, 99%), terbium(III) chloride hexahydrate (TbCl_3_.6H_2_O, 99.9%), and gadolinium(III) chloride hexahydrate (GdCl_3_.6H_2_O, 99.9%) were obtained from Sigma-Aldrich (Saint-Quentin Fallavier, France); Fmoc-L-Lys-OH, Fmoc-L-Lys(Boc)-OH, Fmoc-L-Asp(O*t*Bu)-OH, Fmoc-L-Pro-OH, Fmoc-L-Arg(Pbf)-Wang resin (100–200 mesh) and hexafluorophosphate benzotriazole tetramethyl uronium (HBTU) were purchased from Iris Biotech GmbH (Marktredwitz, Germany). *N*-methyl-2-pyrrolidone (NMP, 99%), *N*-methylmorpholine (NMM, 99%) and triisopropylsilane (TIPS, 98%) were bought from Thermo Scientific Chemicals (formerly Alfa Aesar chemicals) (Karlsruhe, Germany). The singlet oxygen sensor green (SOSG) probe was received from Lumiprobe Company (Maryland, MA, USA). AGuIX Ln NPs ([Ln^3+^] = 50 mM) were provided by NH TherAGuIX Meylan, France).

#### 3.1.2. Materials

Peptides were synthesized with a ResPepXL automated peptide synthesizer (Intavis AG, Bioanalytical Instruments, Köln, Germany).

Compounds were purified by HPLC (Shimadzu LC-10ATvp) with an Agilent Pursuit C18 column, 5 mm column (5 μm, 150 × 21.2 mm), equipped with a UV photodiode array detector (Varian Prostar 335- 190–950 nm) and a spectrofluometric detector (Shimadzu RF–10AXL– 200–650 nm). UV detection was performed at 254 nm and 560 nm. Fluorescence detection at 650 nm was performed after excitation at 415 nm. The HPLC analysis was carried out with the same equipment but with an Agilent Pursuit 5 C18 column (5 μm, 150 × 4.6 mm).

The NMR spectra were recorded on a Brucker Advance 400 spectrophotometer. ^1^H NMR spectra were recorded in DMSO-*d_6_* at 298 K using the solvent residual peak (δ = 2.50 ppm) as an internal reference. Chemical shifts (δ) are expressed in parts per million (ppm), while coupling constants (*J*) are measured in hertz (Hz). The multiplicity is characterized as s for singlet, t for triplet, m for multiplet, H_arom_ for aromatic protons (in RB unit), H_Pyr_ for pyranic protons (in RB unit), and br for broad.

The LC-MS chromatograms were recorded using a Shimadzu brand LCMS-2020 mass spectrometer with a quadrupole (ESI+ electrospray ionization, with detection window of 50 to 200), and coupled to a Shimadzu HPLC chain, LC-20AB pumps, a mini detector with an SPD-M20A diode array and a CTO-20AC oven (Shimadzu, Marne-La-Vallée, France).

Absorption spectra were recorded on a UV-3600 UV–visible double beam spectrophotometer (Shimadzu, Marne-La-Vallée, France). Fluorescence spectra were recorded on a Fluorolog FL3-222 spectrofluorimeter (Horiba Jobin Yvon, Palaiseau, France) equipped with a 450 W Xenon lamp and thermostated cell compartment (25 °C), a UV–visible photomultiplier R928 (Hamamatsu Photonics, Hamamatsu, Japan) and an InGaAs infrared detector (DSS-16A020L Electro-Optical System Inc, Phoenixville, PA, USA). The excitation beam was diffracted by a double ruled grating SPEX monochromator (1200 grooves/mm blazed at 330 nm). The emission beam was diffracted by a double ruled grating SPEX monochromator (1200 grooves/mm blazed at 500 nm). Singlet oxygen emission was detected through a double ruled grating SPEX monochromator (600 grooves/mm blazed at 1 µm) and a long-wave pass (780 nm). All spectra were measured in four-face quartz vials. All the emission spectra (fluorescence and singlet oxygen luminescence) are displayed with the same absorbance (less than 0.2) with the lamp and photomultiplier correction.

The spectral overlap and Förster radius were computed to characterize the energy transfer from the Tb and Gd cation (Tb^3+^, Gd^3+^) to RB. Moreover, the Tb and Gd luminescence decay profile was recorded using a Fluorolog spectrofluorimeter; the excitation wavelength was set at 351 nm for Tb and 273 nm for Gd, and the emission peaks were scanned in the 400–690 nm and 300–600 nm region. The luminescence lifetime of Tb and Gd alone or in mixture with RB was recorded using lifetime Fluorolog. We assessed the 545 nm and 313 nm peak decay as it is the highest Tb and fluorescence peak, respectively. If relevant, we computed the quenching constant (expressed as L.mol^−1^·s^−1^) as Kq = Ksv/τ_0_, where Ksv is the Stern–Volmer constant which was graphically determined; τ_0_ is the Tb fluorescence lifetime without PS.

Irradiations were performed on the OptiRAD platform using a XRAD-320 irradiator (Precision X-rays Inc., Madison, CT, USA). The tube settings were set to 320 kV and 10 mA, and the source-to-surface distance was adjusted to yield a dose rate of 3.0 Gy/min. As was demonstrated in a previous study, a linear relationship exists between the kV X-ray generator setting and scintillator luminescence intensity; therefore, the highest available voltage on the XRAD-320 device (i.e., 320 kV) was used, and subsequently, the current and source-to-surface distance were adjusted to achieve the desired dose rate.

The zeta potential (ξ) for each NPs was determined using Zetasizer Nano-Z (Malvern, UK) equipped with a He-Ne laser at 633 nm.

### 3.2. FRET Experiments

To estimate the FRET ability of a given donor–acceptor FRET pair, the spectral overlap integral (*J_(λ)_*) and Förster radius (*R*_0_) must be calculated using Equations (1) and (2).(1)$$J_{\left(\lambda \right)}=\int F_{D(\lambda )}\varepsilon_{A(\lambda )}\lambda^{4}d\lambda$$
where *J_(λ_*_)_ is the overlap integral (M^−1^·cm^−1^·nm^4^), *λ* is the wavelength (nm), $F_{D(\lambda )}$ is the normalized fluorescence emission of the donor, and $\varepsilon_{A(\lambda )}$ is the extinction coefficient of the acceptor (M^−1^·cm^−1^).(2)$${R_{0}=0.02108\;[\kappa^{2}\Phi_{D}n^{-4}J_{\left(\lambda \right)}]}^{1/6}$$
where *R*_0_ is the Förster radius (nm) at which the efficiency of energy transfer is 50%, *κ*^2^ is the dipole orientation factor taken as 2/3 for a random orientation, *n* is the medium’s refractive index (*n* = 1.3), and *Φ_D_* is the quantum yield of the donor.

The efficiency of energy transfer (*E*) can also be calculated using the simplified Equation (5).(3)$$E=1-\frac{\tau }{\tau_{0}}$$
where *τ*_0_ and *τ* are the fluorescence lifetime of the donor in the absence and presence of the acceptor, respectively,

The energy transfer depends on the amount of quenching in the medium; the quenching constant can be calculated after calculating the Stern–Volmer constant (K_sv_) using Equations (3) and (4).(4)$$\frac{I_{0}}{I}=1+K_{SV}[Q]$$
where *I*_0_ and *I* are the luminescence intensity of the donor in the absence and presence of the acceptor, respectively, *K_sv_* is the Stern–Volmer constant, and [*Q*] is the concentration of the acceptor (i.e., Quencher, Q).(5)$$K_{SV\;}=k_{q\;}\tau_{0}$$
where *k_q_* is the bimolecular rate constant for collisional quenching, and *τ*_0_ is the fluorescence lifetime of the donor in the absence of the acceptor (i.e., Quencher, Q).

### 3.3. Photophysical Experiments

The fluorescence quantum yield (*Φ_f_*) is calculated using Equation (6).(6)$$\Phi_{f}=\Phi_{f0\;}.\;\frac{I_{f}}{I_{f0}}.\frac{{OD}_{0}}{OD}.\;{\left(\frac{n}{n_{0}}\right)}^{2}\;$$
where *Φ_f_* and *Φ_f_*_0_, *I_f_* and *I_f_*_0_, *OD* and *OD*_0_, *n* and *n*_0_ are the quantum yields, fluorescence intensities, optical densities, and refractive indices of the sample and reference, respectively.

Tetraphenylporphyrin (TPP) was chosen as the fluorescence reference standard (*Φ_f_*_0_ = 0.11, toluene) [42].

The ^1^O_2_ production quantum yield (*Φ*_Δ_) is determined using Equation (7).(7)$$\Phi_{\Delta }=\Phi_{\Delta 0\;}.\;\frac{I}{I_{0}}.\frac{{OD}_{0}}{OD}\;$$
where *Φ*_Δ_ and *Φ*_Δ0_, *I* and *I*_0_, *OD* and *OD*_0_ are the quantum yields of ^1^O_2_ production, intensities of ^1^O_2_ production, and optical densities of the sample and reference, respectively.

Eosin Y was chosen as a reference solution (*Φ*_Δ0_ = 0.52, water) [28].

### 3.4. Singlet Oxygen Generation

Singlet oxygen production was evaluated under a range of X-ray doses (5 to 25 Gy at 320 kV/10 mA) with the fluorescent probe SOSG. NP solutions (AGuIX Ln@RB, AGuIX Ln@HA-RB or AGuIX Ln@Mal-K(RB)DKPPR) were used at an RB equivalent concentration of 10 μM. In brief, NPs were mixed in 30 mM Tris/HCl (pH 7.4) containing a 10 μM SOSG probe and X-ray irradiated. Singlet oxygen quenching was achieved by adding NaN_3_ to a final concentration of 10 mM. Fluorescence emission was detected spectroscopically at 525 nm for SOSG after excitation at 473 nm. An optical fiber was inserted in front of the vial containing the reaction mixture to gather emission fluorescence photons. Emission spectra were recorded with a USB2000 spectrometer (Ocean Optics Inc, Dunedin, FL, FWHM = 3.5 nm). The spectrum bandwidth ranged from 340 to 820 nm and the optical fiber was placed across from a transparent vial (Uvette^®^ 220-1600 nm; cat.no. 952010051, Eppendorf, Hamburg, Germany). Home-made software allowed for long acquisition times and synchronization between laser illumination and signal recording. Integration time was set to 100 ms per measurement.

### 3.5. TDA Experiments

TDA experiments were conducted using a TDA-ICP-MS hyphenation between a Sciex P/ACE MDQ instrument and a 7700 Agilent ICP-MS. Fused silica capillaries with an inner diameter of 75 µm and outer diameter of 375 µm, and a total length of 64 cm, were coated with hydroxypropylcellulose (HPC) using a solution of 0.05 g mL^−1^ in water. Detection was carried out by ICP-MS at *m*/*z* = 158 and *m*/*z* = 159 for Gd and Tb detection, respectively. Samples were hydrodynamically injected (0.3psi for 3 s), then mobilized using Tris 10 mM and NaCl 125 mM at 0.7 psi. Between runs, the capillary was flushed at 5 psi for 5 min with the mobilization medium. All measurements were performed at least in duplicates.

The detected peak was fitted by a sum of Gaussian distributions using Origin 8.5 software, according to Equation (8).(8)$$S\left(t\right)=\sum_{i=1}^{2}\left(\frac{A_{i}}{\sigma_{i\;\sqrt{2\pi }}}\right)\exp-\frac{{\left(t-t_{0}\right)}^{2}}{2\sigma_{i}^{2}}$$
where *t*_0_ is the peak residence time, and *σi* and Ai are the area under the curve and the temporal variance associated with each species *i*, respectively.

Under these experimental conditions, the molecular diffusion coefficient *D* is given by(9)$$D=\frac{R_{c}^{2}t_{0}}{24\sigma^{2}}=\frac{{2k}_{B}T}{6\pi \eta D_{h}}$$
where *Rc* is the capillary radius, kB is the Boltzmann constant, *T* is the temperature, *η* is the viscosity, and *D_h_* is the hydrodynamic diameter of the solute.

Therefore, using Equation (9), the hydrodynamic diameters can be calculated from the temporal variances measured from the fitted Taylorgram.

### 3.6. In Vitro Experiments

For the in vitro experiments, human U-251 MG (ECACC 09063001, Salisbury, UK) glioblastoma-derived cells were used.

Roswell Park Memorial Institute medium (RPMI) without phenol red was used to cultivate human U-251 MG (ECACC 09063001, Salisbury, UK) glioblastoma-derived cells. It contained 10% (*v*/*v*) heat-inactivated (30 min at 56 °C) fetal calf serum (Invitrogen, Paisley, UK), 1% (*v*/*v*) non-essential amino acid (Invitrogen), 0.5% (*v*/*v*) essential amino acid (Invitrogen), 1 mM sodium pyruvate (Invitrogen), 1% (*v*/*v*) vitamin (Invitrogen), 0.1 mg/mL L-serine, 0.02 mg/mL L-asparagine (Merck-Sigma), and 1% (*v*/*v*) antibiotics (10,000 U/mL penicillin, 10 mg/mL streptomycin) (Merck-Sigma). Regularly, 10^5^ cells/mL were used to seed the cells, which were then grown at 37 °C in a humidified environment with 5% CO_2_ (Incubator Binder, Tübingen, Germany).

### 3.7. Anchorage-Dependant Clonogenic Assay

The clonogenic assay procedure described in a previous work [40] was used with the following slight changes.

Human U-251 MG glioblastoma cells (ECACC 09063001, Salisbury, UK) were routinely cultivated in Roswell Park Memorial Institute medium (RPMI) without phenol red, containing 10% (*v*/*v*) heat-inactivated (30 min at 56 °C) fetal calf serum (Invitrogen, Paisley, UK), 1% (*v*/*v*) non-essential amino acid (Invitrogen), 0.5% (*v*/*v*) essential amino acid (Invitrogen), 1 mM sodium pyruvate (Invitrogen), 1% (*v*/*v*) vitamin (Invitrogen), 0.1 mg/mL L-serine, 0.02 mg/mL L-asparagine (Merck-Sigma), and 1% (*v*/*v*) antibiotics (10,000 U/mL penicillin, 10 mg/mL streptomycin) (Merck-Sigma). The clonogenic assay procedure has been previously described [31], with the following slight changes. In brief, cells were seeded at 500 cells/well and exposed to NPs with an equivalent concentration of RB (1 µM) for 24 h regardless of the Ln-based AGuIX NPs tested. After incubation, cell layers were washed twice with PBS and X-ray irradiated at 2.0 Gy (320 kV/10 mA). Cells were grown over 7 days. Finally, cell clones were PAF-fixed, stained with crystal violet, and were quantified as previously described [31]. The results are presented by the mean ± SD of triplicates determinations from 3 independent experiments (n = 9). In each experiment, untreated and non-irradiated cells were used as a control. The results obtained were normalized to control cells and analyzed using the Kruskal–Wallis test (with α = 0.05), and post hoc by the Mann–Whitney test (α = 0.05) for unpaired groups and analyzed using the Kruskal–Wallis test (α = 0.05) followed by Dunn’s post hoc analysis (α = 0.05) for unpaired groups.

### 3.8. Affinity to NRP-1

The affinity of AGuIX Ln@Mal-K(RB) and AGuIX Ln@Mal-K(RB)DKPPR to NRP-1 was determined as previously described in terms of IC_50_ in another study [43]. 

## 4. Conclusions

In conclusion, non-radiative FRET is observed for two couples (Tb/RB and Gd/RB) free or chelated in AGuIX NPs. The energy transfer efficiency between Tb/RB is twice that of the Gd/RB pair. The synthesis of RB derivatives (i.e., RB-Ahx, RB-HA, Mal-K(RB), Mal-K(RB)DKPPR) has been successfully completed. One of the advantages of covalently coupling PS to NPs is that the amount to be grafted can be defined to avoid aggregation, which can be detected by spectroscopy. The advantage of using AGuIX is that they possess free amino groups that can be functionalized either directly by creating an amide link or by introducing a thiol group using Traut’s reagent. These bonds are robust and no degradation was observed.

The fluorescence quantum yield of RB derivatives was between 0.10 and 0.15, and very good ^1^O_2_ quantum yields were obtained for PDT of~68% and for X-PDT of ~31%, By exciting AGuIX Ln@RB and the AGuIX Ln@Spacer arm-RB at 558 nm and 351–273 nm, respectively, energy transfer was observed after a delay of 50 µs, with a decrease in the donor fluorescence (Ln = Gd or Tb) and the appearance of the acceptor fluorescence. This energy transfer was confirmed under X-ray irradiation in solution.

The impact of each Ln-based NP was tested on U-251 MG cells. Our results demonstrated that doped NP-RB with Tb is more efficient than those doped with Gd for X-PDT. Similar results were obtained with porphyrin as a PS covalently linked to the NP [40], supporting a better energy transfer to the PS in the experimental conditions tested: 1 µM PS equivalent concentration and cell exposure to 2.0 Gy.

This study shows that every element of the construction process must be taken into account: the type of donor and acceptor, the presence of a spacer arm, etc. Nevertheless, it is difficult to draw a clear conclusion. Indeed, the presence of the HA arm between RB and AGuIX is beneficial in the case of Gd but detrimental in the case of Tb. Concerning the Ahx arm, it is always beneficial. Moreover, it is important to notice that the ^1^O_2_ production in the solution is not necessarily related to the efficiency. Indeed, AGuIX Tb@HA-RB and AGuIX Tb@-RB show the highest ^1^O_2_ production at different doses in Gray (320 kV, 10 mA) but not the best efficacy in vitro. This study is a proof of concept but other studies need to be performed in terms of apoptosis pathways, toxicity to healthy cells, and immunogenicity.

## Data Availability

Data obtained for this study can by obtained under reasonable request.

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
