# Peer review of "Lanthanides-Based Nanoparticles Conjugated with Rose Bengal for FRET-Mediated X-Ray-Induced PDT"

_pharmaceuticals, 2025, doi:10.3390/ph18050672_

Round 1

Reviewer 1 Report

Comments and Suggestions for Authors

Dear Authors,

Thank you for submitting your manuscript, "Lanthanides-based nanoparticles conjugated with Rose Bengal for FRET-mediated X-ray-induced PDT". I am pleased to inform you that your work is potentially interesting and could be valuable to our readers. However, to ensure it meets the standards of our publication, I would like to offer some constructive feedback that I believe will help you improve the manuscript:

Introduction
The introduction outlines PDT but lacks comparisons with other X-PDT systems. The significance of Tb and Gd for glioblastoma therapy is needed.

Methods
The synthesis is detailed, but crucial data on drug-loading efficiency, stability, and release profiles are missing. There is no quantitative assessment of FRET efficiency or clear control descriptions for assays.

Results and Analysis
The SOSG assay points to singlet oxygen production without quantitative benchmarks. Cellular uptake and internalization are not investigated. Anti-cancer efficacy is reported but ignores apoptosis pathways, toxicity to healthy cells, and immunogenicity discussions. Clonogenic results are limited by insufficient replicates.

Figures and Tables
Figures should have higher-resolution images and a comparative table of this system versus existing X-PDT nanoplatforms.

Discussion and Conclusion
Pharmacokinetics, biodistribution, and immunological impacts are not addressed. The limitations section should consider aggregation, stability, and synthesis challenges.

I hope you find this feedback helpful. I encourage you to consider these comments as you revise your manuscript. Thank you again for your submission, and I look forward to reviewing your revised version. 

Best regards.

Comments on the Quality of English Language

It needs improvement. The connection between the paragraphs is missing.

Author Response

Reviewer 1:

Dear Authors,

Thank you for submitting your manuscript, "Lanthanides-based nanoparticles conjugated with Rose Bengal for FRET-mediated X-ray-induced PDT". I am pleased to inform you that your work is potentially interesting and could be valuable to our readers. However, to ensure it meets the standards of our publication, I would like to offer some constructive feedback that I believe will help you improve the manuscript:

Introduction
The introduction outlines PDT but lacks comparisons with other X-PDT systems. The significance of Tb and Gd for glioblastoma therapy is needed.

Thank you for this comment. A paragraph has been added to the introduction to describe PDTX and give the meaning of Tb and Gd.

In this paper, we focus on the use of X-ray,which penetrates deep into the tissues, to excitable nanoparticles (NPs). Indeed, in recent years, photodynamic X-ray excitation therapy (X-PDT), which uses penetrating X-rays as an external excitation source and luminescent X-ray-excited nanoparticles as an energy-transfer medium to indirectly excite the PS, has developed considerably to address the problem of insufficient tissue penetration depth in particular. Recent reviews describe the various nanoparticles used for these purposes (doi: 10.1039/d3ra04984a 10.1016/j.jlumin.2023.119862). The advantage of scintillating materials composed of atoms with a high atomic number (high Z) such as terbium (Tb) or gadolinium (Gd) is that they are able to fluoresce under the excitation of high-energy radiation. This fluorescence is then absorbed by the PS to continue the PDT process (https://doi.org/10.3390/ph17081033).

Methods
The synthesis is detailed, but crucial data on drug-loading efficiency, stability, and release profiles are missing. There is no quantitative assessment of FRET efficiency or clear control descriptions for assays.

Thank you for this comment.

In this study, we did not encapsulate the drug but the PS was covalently coupled to the NP. The grafting was very stable and we did not observe any drug release. To confirm that the nanoparticles were stable, zeta potential measurements were taken regularly.

Concerning FRET efficiency, a table has been added

In short, the FRET energy transfer efficiencies between different couples are calculated according to the equation (3) and presented in table 1.

Table 1 : FRET energy transfer efficiency between AGuIX Tb NPs and RB and AGuIX Gd NPs and RB

Couples

J overloop
(M-1.nm4.cm-1)

R0

(nm)

Type of transfer

Energy transfer efficacity

AGuIX Tb/RB

1.87*1015

3.76

FRET

66%

AGuIX Gd/RB

5.60 *1014

3.08

FRET

27%

As expected from the J overlap, the FRET efficiency is more efficient between AGuIX Tb and RB than between AGuIX Gd and RB.

Results and Analysis

1- The SOSG assay points to singlet oxygen production without quantitative benchmarks.

Thank you

Measuring singlet oxygen directly at 1270 nm in biological is quite impossible. The use of SOSG assay is a standard way of assessing singlet oxygen production. However, it well known that the probe can be self- activated by the excitation light (Koh E, Fluhr R. Singlet oxygen detection in biological systems: Uses and limitations. Plant Signal Behav. 2016 Jul 2;11(7):e1192742. doi: 10.1080/15592324.2016.1192742). That can explain why SOSG signal is still increasing when sodium azide was added to the solution. So having a quantitative measurement of produced singlet oxygen is not feasible in our experimental conditions.

2- Cellular uptake and internalization are not investigated.

Thank you very much.

Indeed, we did not perform any cellular uptake. The aim of this study was to ensure that X-ray excitation of nanoparticles, whether vectorized or not, could destroy cancer cells. In the future, we'll be taking a closer look at cell incorporation for the best candidates.

3- Anti-cancer efficacy is reported but ignores apoptosis pathways, toxicity to healthy cells, and immunogenicity discussions.

Thank you very much.

This paper aimed at demonstrating that the proposed compound could yield a PDT effect. The provided biological results were added not to demonstrate any anti-cancer efficacy but to illustrate whether an effective PDT effect was obtained or not for various constructions. Concerning cell-death mechanisms, PDT approaches have been related to oxidative stress affecting mitochondria to activate caspases pathways for years. However, recent research demonstrated that many pathways may be activated through PDT reaction, including apoptosis, mitotic death or ferroptosis [Mishchenko T et al, Which cell death modality wins the contest for photodynamic therapy of cancer? Cell Death and Disease, 3: 455 (2022)]. In addition, PDT is known to induce the release of damage-Associated Molecular Patterns (DAMP) which will activate dendritic cells as well as CD8+ T lymphocytes and to induce production of pro-inflammatory interleukins [Lerouge L et al,  Targeting Glioblastoma-Associated Macrophages for Photodynamic Therapy Using AGuIX®-Design Nanoparticles, Pharmaceutics. 2023 Mar 20;15(3):997. doi: 10.3390/pharmaceutics15030997].

 Clonogenic results are limited by insufficient replicates. 

The experiments were conducted considering each plate as a sample. The results represented triplicates of 3 independent experiments with n =9 for each condition i.e. each NP tested versus control cells (NP-untreated and not irradiated). It has been added.

Results are the mean ± SD of triplicate determinations from three independent experiments (ie 9 wells/condition).

Figures and Tables

Figures should have higher-resolution images and a comparative table of this system versus existing X-PDT nanoplatforms  

Thank you

We will improve the resolution

Many X-PDT platforms have been developed and it would not be possible to compare our system with all of them.

We added a paragraph

Numerous studies have been carried out to improve PDT-X. Some research is focusing on strategies to enhance luminescence. Methods are also being developed to efficiently load PSs into NPs and improve their solubility in aqueous media. On the other hand, it is important to adjust the distance between donor and acceptor to increase energy transfer efficiency(https://www.researchgate.net/publication/374764448_Xray_excited_luminescent_nanoparticles_for_deep_photodynamic_therapy). Other researchers are trying to find a compromise between the therapeutic index and the ability to treat in depth without increasing toxicity (10.3389/fbioe.2023.1250804). Research indicates that the use of NPs in PDT-X is very promising. (10.3390/nano13040673), especially NPs with rare earths (https://doi.org/10.1016/j.jlumin.2023.119862).

Discussion and Conclusion

Pharmacokinetics, biodistribution, and immunological impacts are not addressed
The reviewer is absolutely right

Indeed, we did not perform any pharmacokinetics, biodistribution and immunology study . The aim of this study was to ensure that X-ray excitation of nanoparticles, whether vectorized or not, could destroy cancer cells. In the future, we'll be taking a closer look at cell incorporation for the best candidates. We added a sentence in the conclusion

This study is a proof of concept but others studies need to be performed in terms of apoptosis pathways, toxicity to healthy cells, and immunogenicity.

The limitations section should consider aggregation, stability, and synthesis challenges.

Thank you for this suggestion

We added a paragraph in the conclusion

One of the advantages of covalently coupling PS to NPs is that the amount to be grafted can be defined to avoid aggregation, which can be detected by spectroscopy. The advantage of using AGuIX is that they possess free amino groups that can be functionalized either directly by creating an amide link or by introducing a thiol group using Traut's reagent. These bonds are robust and no degradation was observed.

I hope you find this feedback helpful. I encourage you to consider these comments as you revise your manuscript. Thank you again for your submission, and I look forward to reviewing your revised version. 

Best regards.

Comments on the Quality of English Language

It needs improvement. The connection between the paragraphs is missing. ( A reviser)

We added some sentences in the document

Energy transfer AGuIX (Tb) or AGuIX (Gd) and RB was first evaluated in solution under light excitation.

Since an energy transfer between AGuIX (Tb) or AGuIX (Gd) and RB was observed in solution under light excitation, RB was covalently couple to Ln-based AGuIX NPs in order to reduce the distance between the lanthanide and the PS to increase FRET efficiency.

We then assessed whether 1O2 generation is achieved under a range of X-ray doses (320 kV/10 mA).

Reviewer 2 Report

Comments and Suggestions for Authors

The paper, entitled “Lanthanides-based nanoparticles conjugated with Rose Bengal for FRET-mediated X-ray-induced PDT”, submitted for review is an interesting article which investigates experimental approaches to evaluate the optical properties of terbium (Tb), gadolinium (Gd) and Rose Bengal (RB) chelated on lanthanide (Ln)-based AGuIX nanoparticles.  This is a nice contribution towards evaluating the materials and their optical features. The authors present interesting experimental results. The article is also well written. My opinion is that the topic is of interest to the readership of Pharmaceutical so the manuscript is publishable, but after minor revision as follows:

  1. In the Introduction section, for better understanding, there should be a diagram (or even two) showing the photodynamic therapy (PDT) mechanism and Dexter and FRET energy transfer mechanism.
  2. The article contains numerous editing errors, which do not diminish its value, but make analysis difficult. It should be carefully reviewed and corrected.

Author Response

Reviewer 2

The paper, entitled “Lanthanides-based nanoparticles conjugated with Rose Bengal for FRET-mediated X-ray-induced PDT”, submitted for review is an interesting article which investigates experimental approaches to evaluate the optical properties of terbium (Tb), gadolinium (Gd) and Rose Bengal (RB) chelated on lanthanide (Ln)-based AGuIX nanoparticles.  This is a nice contribution towards evaluating the materials and their optical features. The authors present interesting experimental results. The article is also well written. My opinion is that the topic is of interest to the readership of Pharmaceutical so the manuscript is publishable, but after minor revision as follows:

Thank you very much

  1. In the Introduction section, for better understanding, there should be a diagram (or even two) showing the photodynamic therapy (PDT) mechanism and Dexter and FRET energy transfer mechanism.

Thank you for this proposition. We added a schemeScheme 1: Energy transfer process between a donor and an acceptor can be radiative or non-radiative (Dexter versus FRET)

  1. The article contains numerous editing errors, which do not diminish its value, but make analysis difficult. It should be carefully reviewed and corrected

Thank you very much. We corrected the manuscript.

Reviewer 3 Report

Comments and Suggestions for Authors

Manuscript titled: “Lanthanides-based nanoparticles conjugated with Rose Bengal for FRET-mediated X-ray-induced PDT” by Batoul Dhaini et al., focuses on the development and optimization of lanthanide-based nanoparticles (AGuIX NPs) conjugated with Rose Bengal (RB) as a photosensitizer (PS) for Förster Resonance Energy Transfer (FRET)-mediated X-ray-induced photodynamic therapy (X-PDT). The goal is to improve the efficiency of PDT for treating deep-seated tumors, which are otherwise limited by the poor penetration of visible light. The optimized nanoparticles exhibit enhanced energy transfer efficiency, singlet oxygen generation, and significant cytotoxicity in glioblastoma cells, highlighting their potential for further in vivo studies and clinical applications.

In my opinion, this research presents a promising approach to overcoming the limitations of conventional PDT using lanthanide-based FRET systems for X-PDT. However, in its current form, the manuscript is poorly written and disorganized, making it difficult to follow. Below are specific comments for improvement:

Major concern:

  1. Important controls are missing to confirm that lanthanides are properly chelated within the AGuIX NPs.

  1. TDA-ICP/MS & DLS Data should not be merely listed in tables but properly discussed and supplemented with actual data.

Minor comments:

Introduction section (p. 2)

The sentence: "Tb, unlike Gd,..." requires clarification—does terbium also function as a therapeutic agent, or is it only discussed in terms of its photophysical properties in the visible spectrum?

The last paragraph begins with: “RB is a water-soluble…”—please elaborate by providing examples of the “interesting photophysical and sonosensitive properties” of RB.

Figures & Tables

Figure 1 – Absorption and emission spectra panels (b and c) should be combined for clarity.

Figure 2 – The resolution is too low; a higher-quality image is needed.

Table 1 & Table 3 – The data for TDA-ICP/MS and DLS experiments should be presented in more detail rather than just summarized in tables.

Results & Discussion section  (p. 3, p. 6)

The sentence: “In the literature, several types of NPs have been linked to RB for PDT, such as silica NPs, organic NPs, nanogels, nanocomplexes, hybrid NPs, MOFs. The covalent coupling of RB to NPs is considered more efficient than encapsulation.”—this statement requires additional citations to support it.

The last paragraph on p. 3 is redundant, as it repeats information from the first paragraph on p. 2.

  1. 6, paragraph starting with "Figure 5 shows…"—this section does not contribute significantly to the discussion and should be removed.

Materials & Methods section

  • The organization needs improvement. Suggestions:

Section 4.2 ("Synthesis") – Move to the Supporting Information section.

Section 4.7.1 ("Anchorage-dependent clonogenic assay") – Clarify how long the cells were treated with RB and Ln-based AGuIX NPs. Were any negative controls included?

Section 4.5 ("X-ray experiments") – The text is in italics; is this intentional? 

Section "Materials" – Needs clearer subdivisions (e.g., NMR Studies, UV-Vis Studies, etc.).

DLS experiment is missing explicit description.

Conclusion section

The conclusion is too technical and lacks a clear statement of impact.

The first paragraph of the Conclusion belongs in the Introduction.

The first sentence ending with “….solution 1” needs to be restructured for clarity

Author Response

Reviewer 3:

Comments and Suggestions for Authors

Manuscript titled: “Lanthanides-based nanoparticles conjugated with Rose Bengal for FRET-mediated X-ray-induced PDT” by Batoul Dhaini et al., focuses on the development and optimization of lanthanide-based nanoparticles (AGuIX NPs) conjugated with Rose Bengal (RB) as a photosensitizer (PS) for Förster Resonance Energy Transfer (FRET)-mediated X-ray-induced photodynamic therapy (X-PDT). The goal is to improve the efficiency of PDT for treating deep-seated tumors, which are otherwise limited by the poor penetration of visible light. The optimized nanoparticles exhibit enhanced energy transfer efficiency, singlet oxygen generation, and significant cytotoxicity in glioblastoma cells, highlighting their potential for further in vivo studies and clinical applications.

In my opinion, this research presents a promising approach to overcoming the limitations of conventional PDT using lanthanide-based FRET systems for X-PDT. However, in its current form, the manuscript is poorly written and disorganized, making it difficult to follow. Below are specific comments for improvement:

Major concern:

  1. Important controls are missing to confirm that lanthanides are properly chelated within the AGuIX NPs.

AGuIX nanoparticles were obtained from NH TherAguix and all characterizations were carried out by the company.

  1. TDA-ICP/MS & DLS Data should not be merely listed in tables but properly discussed and supplemented with actual data.

Thank you for this suggestion

A paragraph has been added

TDA experiments were conducted using a TDA-ICP-MS hyphenation between a Sciex P/ACE MDQ instrument and a 7700 Agilent ICP-MS. Fused silica capillaries with an inner diameter of 75 µm and outer diameter of 375 µm, and a total length of 64 cm, were coated with hydroxypropylcellulose (HPC) using a solution of 0.05 g mL-1 in water. Detection was carried out by ICP-MS at m/z=158 and m/z=159, for Gd and Tb detection respectively. Samples were hydrodynamically injected (0.3psi for 3 s), then mobilized using Tris 10 mM, NaCl 125 mM at 0.7 psi. Between runs, the capillary was flushed at 5 psi for 5 min with the mobilization medium. All measurements were performed at least in duplicates.

The detected peak was then fitted by a sum of Gaussian distributions using Origin 8.5 software, according to the following equation:

                                 equation 1

where t0 is the peak residence time, and σi and Ai are the area under the curve and the temporal variance associated to each species i respectively.

Under these experimental conditions, the molecular diffusion coefficient D is given by

                                     equation 2

where Rc is the capillary radius, kB is the Boltzmann constant, T is the temperature, η is the viscosity, and Dh is the hydrodynamic diameter of the solute.

Therefore, using equation 2, the hydrodynamic diameters can be calculated from the temporal variances measured from the fitted Taylorgram.

Minor comments:

Introduction section (p. 2)

The sentence: "Tb, unlike Gd,..." requires clarification—does terbium also function as a therapeutic agent, or is it only discussed in terms of its photophysical properties in the visible spectrum?

Thank you

This formulation is only to discuss in terms of photophysical properties (the difference in emission between the two lanthanides). There is no therapeutic role for Tb and Gd

The last paragraph begins with: “RB is a water-soluble…”—please elaborate by providing examples of the “interesting photophysical and sonosensitive properties” of RB.

Thank you

We added a paragraph

For example, Nonaka et al. (Nonaka M, Yamamoto M, Yoshino SI, Umemura SI, Sa- saki K, Fukushima T. Sonodynamic therapy consisting of focused ultrasound and a photosensitizer causes a selective antitumor effect in a rat intracranial glioma model. Antic- ancer Res 2009;29:943–950.) applied sonodynamic therapy (SDT) with RB and focused ultra-sound to experimental intracranial glioma in rats. A further application of sono-activated RB was reported by Nakonechny et al. (Nakonechny F, Nisnevitch M, Nitzan Y, Nisnevitch M. Sonodynamic excitation of rose bengal for eradication of gram-positive and gram-negative bacteria sonodynamic excitation of rose bengal for eradication of gram-positive and gram-negative bacteria. Biomed Res Int 2013;2013: 684930.) who demonstrated it was possible to eradicate gram-positive and gram-negative bacteriaby applying SDT using activated RB in vitro. RB photo-activation can be used for external application on the body, such as wound sealing or corneal crosslinking, whereas the use of sono-activated RB could be further explored for cancer treatment.

Figures & Tables

Figure 1 – Absorption and emission spectra panels (b and c) should be combined for clarity.

This has been done.

Figure 2 – The resolution is too low; a higher-quality image is needed.

We are aware resolution is too low and we will improve it.

Table 1 & Table 3 – The data for TDA-ICP/MS and DLS experiments should be presented in more detail rather than just summarized in tables. 

Thank you very much

We added a paragraph and change the table

As can be seen for Tb-based NPs, the TDA analysis shows that AGuIX Tb @RB, AGuIX Tb@HA-RB and AGuIX Tb@Ahx-RB exhibit a higher hydrodynamic diameter, which reflects the efficiency of the coupling. Moreover, the analysis reveals the presence of 2 populations in these samples, with a main population which can be attributed to the functionalized NP, while the other remaining might be attributed to a slight hydrolysis of the NP (see Supporting Information).

Table 2: Zeta potential (ζ) and size values for AGuIX Ln, AGuIX Ln@RB and AGuIX Ln@spacer arm-RB

* Values between brackets indicate the percentage of Ln involved in each species

Samples

ζ (mV)

Size* (nm)

ζ (mV)

Size* (nm)

Ln = Tb

Ln = Gd

AGuIX Ln

+ 7

5.6 ± 0.1 ((95±3)%)

+ 1

1.8 ± 0.1 (100%)

AGuIX Ln@RB

- 16

3.3 ± 0.3 ((23±5)%)

7.0 ± 0.2 ((77±5)%)

- 11

2.6 ± 0.1 ((90±7)%)

AGuIX Ln@HA-RB

- 27

3.8 ± 0.2 ((33±4)%)

7.3 ± 0.2 ((67±4)%)

- 17

2.9 ± 0.1 ((97±2)%)

AGuIX Ln@Ahx-RB

- 30

3.0 ± 0.2 ((26±4)%)

7.3 ± 0.1 ((74±4)%)

- 25

2.7 ± 0.1 ((95±3)%)

Results & Discussion section  (p. 3, p. 6)

The sentence: “In the literature, several types of NPs have been linked to RB for PDT, such as silica NPs [49] , organic NPs 65, nanogels 103, nanocomplexes 101, hybrid NPs 94, MOFs 118. The covalent coupling of RB to NPs is considered more efficient than encapsulation.”—this statement requires additional citations to support it.

Thank you very much, a paragraph has been added

In the literature, several types of NPs have been linked to RB for PDT, such as silica NPs  (Uppal, A.; Jain, B.; Gupta, P.K.; Das, K. Photodynamic Action of Rose Bengal Silica Nanoparticle Complex on Breast and Oral Cancer Cell Lines. Photochem. Photobiol. 2011, 87, 1146–1151)., organic NPs), nanogels (Torres-Martínez, A.; Bedrina, B.; Falomir, E.; Marín, M.J.; Angulo-Pachón, C.A.; Galindo, F.; Miravet, J.F. Non-Polymeric Nanogels as Versatile Nanocarriers: Intracellular Transport of the Photosensitizers Rose Bengal and Hypericin for Photodynamic Therapy. ACS Appl. Bio Mater. 2021, 4, 3658–3669), nanocomplexes (Su, P.; Zhu, Z.; Fan, Q.; Cao, J.; Wang, Y.; Yang, X.; Cheng, B.; Liu, W.; Tang, Y. Surface ligand coordination induced self-assembly of a nanohybrid for efficient photodynamic therapy and imaging. Inorg. Chem. Front. 2018, 5, 2620–262), hybrid NPs  (Cao, H.; Qi, Y.; Yang, Y.; Wang, L.; Sun, J.; Li, Y.; Xia, J.; Wang, H.; Li, A.P.D.J. Assembled Nanocomplex for Improving Photodynamic Therapy through Intraparticle Fluorescence Resonance Energy Transfer. Chem. Asian J. 2018, 13, 3540–3546), MOFs (Zhao, X.; Li, Y.; Du, L.; Deng, Z.; Jiang, M.; Zeng, S. Soft X-ray Stimulated Lanthanide@MOF Nanoprobe for Amplifying Deep Tissue Synergistic Photodynamic and Antitumor Immunotherapy. Adv. Health Mater. 2021, 10, 2101174) The covalent coupling of RB to NPs is considered more efficient than encapsulation 2.

The last paragraph on p. 3 is redundant, as it repeats information from the first paragraph on p. 2.

Thank you very much, we removed some sentence in the first paragraph on p. 2.

  1. 6, paragraph starting with "Figure 5 shows…"—this section does not contribute significantly to the discussion and should be removed.

This paragraph has not been removed but considerably reduced

The decrease in luminescence intensity and lifetime for Tb and Gd is consistent with non-radiative energy transfer. The Förster radius R0 of 4.33 and 2.73 nm for Tb and Gd, respectively, is < 10 nm and confirms that the non-radiative transfer is FRET-like.

The exponential decrease in the luminescence lifetime and intensity of Tb/Gd, respectively, confirms that the non-radiative FRET transfer could be due to dynamic and static inhibitions with, for the TbCl3/RB pair, kq = 5.5*108 M-1.s-1 and Ksv = 22.3*104 M-1 (slope of the curve τ0/τ = f([RB]) while τ0 = 398 μs). For the GdCl3/RB pair, it was calculated Ksv = 9.6*104 M-1, kq = 2.3*108 M-1.s-1 with τ0 = 410 μs.

It is possible to compare the two GdCl3/RB and TbCl3/RB pairs. The spectral overlap between Tb and RB is 10 times higher than that of Gd and RB, leading to better energy transfer for the TbCl3/RB pair than that of GdCl3/RB. This result is confirmed by the fact that R0 for TbCl3/RB (4.33 nm) is larger than for GdCl3/RB (2.73 nm), which means that at a distance of 4.33 nm the efficiency of FRET is 50% while this distance must be shortened at 2.73 nm for the GdCl3/RB couple. It is possible to evaluate, according to Equation 3, the simplified energy transfer efficiency (E) using the luminescence lifetimes without and with quenching. E is approximatly 65% for the TbCl3/RB pair, and 42% for the GdCl3/RB pair.

Additionally, it must be taken into account that in water the luminescence lifetime of trivalent Ln changes depending on solvation 27. To better understand the effect of this phenomenon on the luminescence lifetime of Ln, an evaluation of the number of water molecules in the first coordination sphere was carried out.

Materials & Methods section

  • The organization needs improvement. Suggestions:

Section 4.2 ("Synthesis") – Move to the Supporting Information section.

Thank you for this suggestion

The section “Synthesis” has been moved to Supporting Information section.

Section 4.7.1 ("Anchorage-dependent clonogenic assay") – Clarify how long the cells were treated with RB and Ln-based AGuIX NPs. Were any negative controls included?

Thank you. Cells were exposed to NPs for 24 h. The description has been changed.

Human U-251 MG glioblastoma cells (ECACC 09063001, Salisbury, UK) were routinely cultivated in Roswell Park Memorial Institute medium (RPMI) without phenol red, contained 10% (v/v) heat-inactivated (30 min at 56 °C) fetal calf serum (Invitrogen, Paisley, UK), 1% (v/v) non-essential amino acid (Invitrogen), 0.5% (v/v) essential amino acid (Invitrogen), 1 mM sodium pyruvate (Invitrogen), 1% (v/v) vitamin (Invitrogen), 0.1 mg/mL L-serine, 0.02 mg/mL L-asparagine (Merck-Sigma), and 1% (v/v) antibiotics (10,000 U/mL penicillin, 10 mg/mL streptomycin) (Merck-Sigma). Clonogenic assay procedure was previously described31 , with following slight changes. In brief, cells were seeded at 500 cells/well and exposed to NPs with an equivalent concentration of RB (1 µM) for 24 h whatever the Ln-based AGuIX NPs tested. After incubation, cell layers were washed twice with PBS and X-ray irradiated at 2.0 Gy (320 kV/10 mA). Cells were let grown over 7 days. Finally, cell clones were PAF-fixed, stained with cristal violet and they were quantified as previously described31 . Results are presented by the mean ± SD of triplicates determinations from 3 independent experiments (n = 9). In each experiment, untreated and non irradiated cells were used as a control. The results obtaind were normalized to control cells and analysed using the Kruskal-Wallis test (with α = 0.05), and post-hoc by the Mann-Whitney test (α = 0.05) for unpaired groups and analysed using the Kruskal-Wallis test (α = 0.05) followed by Dunn post-hoc analysis (α = 0.05) for unpaired groups.

Section 4.5 ("X-ray experiments") – The text is in italics; is this intentional? 

It has been changed

Section "Materials" – Needs clearer subdivisions (e.g., NMR Studies, UV-Vis Studies, etc.).

It has been changed

DLS experiment is missing explicit description.

A paragraph has been added

4.5. TDA experiments

TDA experiments were conducted using a TDA-ICP-MS hyphenation between a Sciex P/ACE MDQ instrument and a 7700 Agilent ICP-MS. Fused silica capillaries with an inner diameter of 75 µm and outer diameter of 375 µm, and a total length of 64 cm, were coated with hydroxypropylcellulose (HPC) using a solution of 0.05 g mL-1 in water. Detection was carried out by ICP-MS at m/z=158 and m/z=159, for Gd and Tb detection respectively. Samples were hydrodynamically injected (0.3psi for 3 s), then mobilized using Tris 10 mM, NaCl 125 mM at 0.7 psi. Between runs, the capillary was flushed at 5 psi for 5 min with the mobilization medium. All measurements were performed at least in duplicates.

The detected peak was fitted by a sum of Gaussian distributions using Origin 8.5 software, according to Equation (8).

                                                                                                     (8)

where t0 is the peak residence time, and σi and Ai are the area under the curve and the temporal variance associated to each species i, respectively.

Under these experimental conditions, the molecular diffusion coefficient D is given by

                                                                                                                                   (9)

where Rc is the capillary radius, kB is the Boltzmann constant, T is the temperature, η is the viscosity, and Dh is the hydrodynamic diameter of the solute.

Therefore, using equation 9, the hydrodynamic diameters can be calculated from the temporal variances measured from the fitted Taylorgram.

Conclusion section

The conclusion is too technical and lacks a clear statement of impact.

The first paragraph of the Conclusion belongs in the Introduction.

The first sentence ending with “….solution 1” needs to be restructured for clarity

Thank you very much, we changed the conclusion

Recently, we demonstrated the possibility of obtaining energy transfer after excitation by light or X-ray between Gd or Tb and the PS Photofrin in solution 1. We have also demonstrated that i) this energy transfer occurs when Photofrin is adsorbed onto the AGuIX NPs, leading to the production of 1O2., and ii) energy transfer occurs when a the tetraphenyl monocarboxylic porphyrin (P1) is covalently coupled to AGuIX 30. AGuIX coupled to RB shows better results than AGuIX coupled to P1 30. Indeed, the number of normalized living cells for AGuIX Tb@HA-RB is 0.48% for 1 µM of RB at 2 Gy, compared with 0.515% for AGuIX Tb@P1.

In conclusion, non-radiative FRET energy transfer is observed for two couples (Tb/RB and Gd/RB) free or chelated in AGuIX NPs. FRET efficiency is influenced by the spectral overlap between the UV-visible absorption spectra of RB and the luminescence emission spectra of Ln, which in turn influences the Förster radius (R0). Given that the Kq value of the Tb/RB pair is twice that of the Gd/RB pair, The energy transfer efficiency between Tb/RB is twice that of the Gd/RB pair. The synthesis of RB derivatives (i.e., RB-Ahx, RB-HA, Mal-K(RB), Mal-K(RB)DKPPR) has been successfully completed. Photophysical studies of RB precursors show that RB coupling reduces its molar extinction coefficient, fluorescence and 1O2 quantum yields. These RB derivatives were coupled to AGuIX Ln NPs. The photophysical properties of RB were recorded and for both types of NPs (Ln = Gd and Tb), an increase in the fluorescence lifetime of RB was observed when coupled to NPs. The fluorescence quantum yield of RB derivatives are between 0.10 and 0.15, and very good 1O2 quantum yields were obtained ~68 % for PDT and ~31 % for X-PDT. By exciting AGuIX Ln@RB and AGuIX Ln@Spacer arm-RB at 558 nm and 351-273 nm, respectively, energy transfer was observed after a delay of 50 µs, with the decrease in the donor fluorescence (Ln = Gd or Tb) and the appearance of the acceptor fluorescence. This energy transfer was confirmed under X-ray irradiation in solution.

The impact of each Ln-based NPs was tested on U-251 MG cells. Our results demonstrated that doped NP-RB with Tb is more efficient than those doped with Gd for X-PDT. Similar results were obtained with porphyrin as a PS covalently linked to the NP 31 supporting a better energy transfer to the PS in the experimental conditions tested, 1 µM PS equivalent concentration and cell exposure to 2.0 Gy.

This study shows that every element of the construction process must be taken into account: the type of donor and acceptor, the presence of a spacer arm, etc. Neverthelesse, it is difficult to draw a clear conclusion. Indeed, the presence of the HA arm between RB and AGuIX is beneficial in the case of Gd but detrimental in the case of Tb. Concerning Ahx arm, it is always beneficial. Moreover, it is important to notice that the 1O2 production in solution is not necessarily related to the efficiency. Indeed, AGuIX Tb@HA-RB and AGuIX Tb@-RB show the highest 1O2 production at different doses in Gray (320 kV, 10 mA) but not the best efficacy in vitro. This study is a proof of concept but others studies need to be performed in terms of apoptosis pathways, toxicity to healthy cells, and immunogenicity.

Round 2

Reviewer 3 Report

Comments and Suggestions for Authors

All comments were addressed accordingly.